# Soil Metabolomics Predict Microbial Taxa as Biomarkers of Moisture Status in Soils from a Tidal Wetland

**DOI:** 10.3390/microorganisms10081653

**Published:** 2022-08-16

**Authors:** Taniya RoyChowdhury, Lisa M. Bramer, Joseph Brown, Young-Mo Kim, Erika Zink, Thomas O. Metz, Lee Ann McCue, Heida L. Diefenderfer, Vanessa Bailey

**Affiliations:** 1Earth and Biological Sciences Directorate, Pacific Northwest National Laboratory, Richland, WA 99352, USA; 2Energy and Environment Directorate, Pacific Northwest National Laboratory, Sequim, WA 98382, USA

**Keywords:** 16S rRNA gene, soil metabolomics, random forest classification, wetland microbiome

## Abstract

We present observations from a laboratory-controlled study on the impacts of extreme wetting and drying on a wetland soil microbiome. Our approach was to experimentally challenge the soil microbiome to understand impacts on anaerobic carbon cycling processes as the system transitions from dryness to saturation and vice-versa. Specifically, we tested for impacts on stress responses related to shifts from wet to drought conditions. We used a combination of high-resolution data for small organic chemical compounds (metabolites) and biological (community structure based on 16S rRNA gene sequencing) features. Using a robust correlation-independent data approach, we further tested the predictive power of soil metabolites for the presence or absence of taxa. Here, we demonstrate that taking an untargeted, multidimensional data approach to the interpretation of metabolomics has the potential to indicate the causative pathways selecting for the observed bacterial community structure in soils.

## 1. Introduction

Hydrologic fluctuations are a driving factor regulating wetland biogeochemical processes, but controlled studies investigating the effect of rapid and extreme moisture regime shifts on the soil microbiome, specifically on anaerobic carbon (C) cycling pathways, are limited. The few studies on the impact of fluctuating precipitation regimes (extreme wetting and drying) on the soil microbiome have been carried out on upland systems either by characterizing microbial community composition shifts [1,2,3,4,5], or by identifying correlative associations between the microbial community and metabolite data [6,7]. The latter correlations are useful for describing, predicting, and managing functional traits of the microbiome [8] and can inform new hypotheses regarding microbe-dependent ecosystem processes [9,10]. For example, we know that there are direct links between soil redox conditions and the soil microbiome that impact biogeochemical functions including greenhouse gas production, iron reduction, and C-sequestration potentials [11,12,13,14]. Still, metrics such as correlations are limited in detecting linear associations. To uncover causal relationships, statistical approaches free of the assumption of linear relationships, as is commonly seen in multidimensional microbiome datasets from soil, must be utilized. Such approaches that link community membership information (e.g., 16S rRNA gene surveys) with microbial metabolic processes remain unexplored in soil systems.

Untargeted mass spectrometry approaches to characterize soil/sediment metabolomes have only recently been developed and applied [3,15,16]. From these techniques, metabolic pathways have been inferred using network analysis methods [7] or through integration of gene expression (metatranscriptomes) with metabolite profiles using association networks [3]. The rationale for the use of association networks to analyze metabolomics data is that differences or commonalities in biological processes are reflected in the characteristics of the inferred correlation networks [17]. Generally, the correlations observed in metabolomics data are the result of the combination of all reactions and regulatory processes in the network; there may be no correlations between metabolites that are close in a certain metabolic pathway [18]. Pairwise correlations are also limited because only pairs of features can be assessed at once and correlation quantifies the linear relationship between two features. Such approaches allow us to visualize and examine correlations/relationships between pairs of features simultaneously but may not work for soil metabolomic profiling that are typically characterized by complex multivariate relationships with the microbial community structure. Inferring soil functions from correlations between observed features for microbial taxa and small organic molecules (metabolite and/or substrate) is inadequate and provides limited insight into relationships between these features. 

An area of soil multi-omics research needing major advances is in the application of metabolome data from untargeted approaches, i.e., mass spectrometry, to infer microbial community composition and predict community metabolism. Derivatives of the relative abundance of a set of Operational Taxonomic Units (OTUs) from soil microbiomes have been used to infer ecological modules or clusters [19,20]. However, microbial taxa composition alone is a poor predictor of soil function because it does not link ecology to phenotype, i.e., the plethora of physiological adaptations expressed in prokaryotes, such as the use of different C functional groups, and terminal electron acceptors as a soil environment change. Here, we adopted the definition of the metabolome as an effective phenotype modulator [21] and used it to infer soil microbiome biomarkers under altered soil moisture conditions as described below. 

We used controlled laboratory incubations to simulate the following three soil moisture conditions: constant saturation, fluctuating, and drying as previously described [22]. Our objective was to force shifts in the soil microbial community composition and metabolite profiles. We hypothesized that the accumulation of metabolites central to fermentative pathways, and anaerobic respiration under wet/saturated conditions would provide evidence for the presence of microbial taxa specializing in these pathways. Similarly, microbial taxa resilient to rapid drying would be predicted by metabolites indicative of drought/stress tolerance. We applied a data-integration approach to determine if and how the composition and abundance of metabolites predict the presence/absence of microbial taxa in soils. We used a random forest classifier [23] to integrate data from 16S rRNA sequencing and metabolite profiles from gas chromatography–mass spectrometry (GC–MS). Our findings indicated that the use of metabolites as predictors of presence/absences of taxa using random forest models can be a meaningful approach to link community structure to function.

## 2. Materials and Methods

### 2.1. Soil Collection and Incubation Conditions

#### 2.1.1. Site

The study site is located within the tidally influenced, freshwater Secret River wetland on the Columbia River estuary floodplain in southwestern Washington State, USA. In a recent synoptic study of wetlands in the Pacific Northwest USA, the tidal forested wetland at Secret River was shown to have one of the highest total ecosystem C stocks in the world, exceeding mangroves, and more C was sequestered in the soil (to 3-m depth) than in aboveground biomass (Kauffman et al., 2020 [24]). Further details about the study site and soil core collection have been previously described [22]. 

Fifteen soil cores were sampled in opaque polyvinyl chloride columns (5 cm height × 2.5-cm i.d.) in 2016 using a grid sampling technique [22] and transported on ice to the Pacific Northwest National Laboratory, Richland, WA, USA, where they were stored at 4 °C until further use. Cores were pre-incubated at 20 °C for 18 days under field-moist conditions (~50% water *w*/*w*) and monitored daily for greenhouse gas production, i.e., carbon dioxide (CO_2_) and methane (CH_4_), by gas chromatography. At the end of the pre-incubation period, cores were randomly assigned to each of the three treatments: Wet-Dry (W-D, n = 5), Dry-Wet (D-W, n = 4), and continuous saturation (Sat, n = 3). At least 3 replicates per treatment were deemed minimum for the experiment; additional replicates for the W-D and D-W treatments were included to perform statistical power analyses to determine the minimum replication to confidently attribute sources of variance observed for bulk process measurements (soil respiration, previously published [22]) and molecular analysis, i.e., 16S rRNA sequencing and metabolomics, presented in this study. Three of the 15 cores collected were used for additional soil property analysis [22].

#### 2.1.2. Soil Incubation

Samples were maintained in a humidity-controlled (60% relative humidity) chamber for 14 days following treatment initiation and thereafter transferred to a biosafety cabinet at 25 °C for the duration of the incubation (120 days). It took up to 16 h for soil cores to be saturated by placing them in standing water after which an additional 1–2 mL of water was added to the soil column surface to limit evaporative losses. The continuously saturated (Sat) samples were saturated as described above, maintained at constant weights by addition of 1–2 mL of water and incubated anoxically (headspace purged with 99.9% N_2_). The W-D treatment was saturated and initially incubated under a microaerophilic headspace (purged with 70% N_2_) for 14 days and subsequently maintained anoxically for the next 25 days (wet phase). For the D-W treatment, pre-incubated field-moist samples were allowed to dry to a constant weight by evaporation under a microaerophilic headspace for 14 days and then maintained under aerobic conditions for the next 25 days of the dry phase at 20 °C.

After 39 days from treatment onset, the moisture and headspace conditions for the W-D and D-W treatments were reversed: the W-D samples initially under wet/anoxic conditions were subjected to drying by evaporation under aerobic conditions at 20 °C (moisture data published in [22]). The D-W samples from the initially dry treatments were wetted by addition of de-gassed sterile de-ionized water until the rate of evaporative loss was lower than the volume of water added and incubated under an anoxic headspace. Wetting was conducted in the same manner as the initial wetting treatments, described above. Gas measurements for headspace CO_2_ and CH_4_ were measured throughout the incubation and have been previously reported [22].

#### 2.1.3. Soil Core Sampling and Analyses

All sample handling and post-incubation processing for the D-W and Sat samples were conducted inside an anoxic glove bag (Coy Laboratories, O_2_ < 5 ppm, 2.1% H_2_): soil cores were extruded from the PVC columns, homogenized, and subsampled for deoxyribonucleic acid (DNA) and extracellular metabolite extraction analyses (described below). DNA was extracted immediately on the last day of incubation. DNA was stored at −80 °C until further analysis. Soil subsamples were preserved at −20 °C until metabolite extraction. The experiment generated a total of 60 samples for sequence and metabolite analysis.

### 2.2. DNA Extraction and 16S rRNA Sequence Analysis

DNA was extracted from 0.25 g portions (five technical replicates per core) of homogenized subsamples using the PowerSoil DNA isolation kit (MoBio Laboratories Inc., Carlsbad, CA, USA) according to the manufacturer’s instructions. DNA extracts were quantified using the Qubit Fluorometer 2.0 (Thermo Fisher Scientific Inc., Waltham, MA, USA) and checked for quality (A260/A280 > 1.9) using the NanoDrop 2000 (Thermo Fisher Scientific Inc.). Additional DNA purification was performed using the ZR-96 Genomic DNA Clean and Concentrator-5 kit (Zymo Research Corporation, Irvine, CA, USA). The V4 region of the 16S rRNA gene was amplified using the PCR protocol developed by the Earth Microbiome Project [25]; this protocol was modified to include a 12-base barcode sequence in the forward primer. Amplicons were sequenced on an Illumina MiSeq using the 500 cycle MiSeq Reagent Kit v2 (http://www.illumina.com/ (accessed on 16 September 2016)) according to manufacturer’s instructions.

Raw sequence reads were de-multiplexed using *ea-utils* [26] with zero mismatches allowed in the barcode sequence. Reads were quality-filtered with BBDuk2 [27] to remove adapter sequences and PhiX with a matching kmer length of 31 bp at a hamming distance of 1. Reads shorter than 51 bp were discarded. The forward and reverse reads were merged using USEARCH [28] with a minimum length threshold of 175 bp and maximum mismatch of 1%. Sequences were de-replicated (minimum sequence abundance of 2) and clustered using the distance-based, “greedy clustering” method (i.e., maximize similarity within clusters, minimize similarity between clusters) of USEARCH at 97% pairwise sequence identity among OTU member sequences. De novo prediction of chimeric sequences was performed using USEARCH during clustering. Taxonomy was assigned to OTU sequences at a minimum identity cutoff fraction of 0.8 using the global alignment method implemented in USEARCH across RDP Trainset database version 15 trained with the UTAX 250 bp configuration [29]. OTU seed sequences were filtered against RDP Gold reference database version 9 to identify chimeric OTUs using USEARCH. 

### 2.3. Metabolome Extraction and GC–MS Analysis

Extracellular metabolites were extracted from 2 g of wet soil from each core (approx. 1–1.5 g dry weight equivalent) (with five technical replicates) using 10 mL of de-ionized water. Samples were shaken on a mechanical shaker for 12 h at 4 °C, centrifuged at 2500× *g* for 15 min at 4 °C, and filtered through 0.2 µm polyethersulfone membrane filters (Pall Corporation, New York, NY, USA). The polar metabolites were completely dried under a speed vacuum concentrator, then chemically derivatized and analyzed by gas chromatography–mass spectrometry (GC–MS). An Agilent 7890A gas chromatograph coupled with a quadrupole 5975C mass spectrometer (Agilent Technologies, Inc., Santa Clara, CA, USA) was used for all analyses. Metabolites were derivatized as previously described [3,30] by adding 20 µL of methoxyamine solution (30 mg mL^−1^ in pyridine) and were incubated at 37 °C for 90 min to protect the carbonyl groups and reduce carbohydrate isoforms. Then, 80 µL of *N*-methyl-*N*-(trimethylsilyl)-trifluoroacetamide with 1% trimethylchlorosilane were added to each sample to trimethylsilyate the hydroxyl and amine groups for 30 min. The samples were cooled to room temperature prior to GC–MS analysis. Data collected by GC–MS were processed using the MetaboliteDetector software, version 2.5 beta [31]. Retention indices of detected metabolites were calculated based on analysis of the fatty acid methyl esters mixture (C8–C28), followed by chromatographic alignment across all analyses after deconvolution. Metabolites were initially identified by matching experimental spectra to a PNNL-augmented version of the Agilent Fiehn Metabolomics Library, containing spectra and validated retention indices for over 900 metabolites [32]. Additionally, metabolites were cross-checked by matching with NIST17 GC-MS Spectral Library. All metabolite identifications were manually validated to minimize deconvolution and identification errors during the automated data processing.

## 3. Data Analyses

### 3.1. 16S rRNA Sequence Data Filtering

The number of samples that successfully sequenced for each treatment were: 10 (out of 20) for D-W, 20 (out of 25) for W-D, and 13 (out of 15) for Sat (uneven replicates were the result of limits on soil core collection and sequencing failures). A total of 14,999 OTUs were observed in at least one of the 43 samples. An outlier detected using a randomization test based on pairwise Jaccard distances was removed as was a sample without corresponding metabolite data. Additionally, any OTUs that were observed in less than 2 samples were removed. A total of 4191 such OTUs were removed, leaving 12,649 OTUs in the dataset of 41 samples.

Rarefaction curves for 16S rRNA data for each sample were generated using the R package *vegan* [33], using R programming language (R Core Team, 2018), to assess the sufficiency of depth of sequencing to calculate species richness and inform outlier removal (Appendix A). OTU tables were generated using the *phyloseq* R package [34]. OTUs present were counted after removal of singletons for all downstream analyses. Count data were screened for changes in relative abundance based on upper 75th-quartile normalized library counts (Bullard et al., 2010 [35]). A significant change in relative abundance was considered at *p-adj.* < 0.05. Significance values were corrected for multiple tests using the Benjamini–Hochberg procedure (Dalmasso et al., 2005 [36]). A nonmetric multidimensional scaling (NMDS) plot with all OTUs was used to visualize the data based on Bray–Curtis distance metrics. NMDS was performed using *vegan* package v.2.5.4 [33]. Figures were rendered using *ggplot2* v.3.1.1.

### 3.2. Metabolomics Data Univariate Analysis

A total of 125 metabolites were observed in at least one of the 41 samples. Any metabolites that did not have enough data to conduct quantitative statistical testing (observed in 2 samples for at least 2 of the 3 treatments) were removed and missing observations were replaced with NAs. The data were log_2_ transformed and then normalized via median centering as previously reported [3]. For each metabolite, a mixed-effects linear model was fit to the data assuming a conditional normal distribution. The fixed effect was “moisture”, and a random effect for the soil core was included using *lme4* v1.1-19. All pairwise group comparisons were conducted, and a one-step multiple test adjustment was made to the resulting *p*-values to obtain the adjusted *p*-values using *multcomp* v1.4-10.

### 3.3. Metabolite Data Processing for Random Forest Classification Approach 

A total of 125 metabolites were observed in at least one of the 41 samples. Because random forest classifiers require complete observations, all missing values were imputed by replacing the missing value with half of the lowest observed log_2_ metabolite abundance across all the data (3.2992124). The data were then normalized via median centering. The number of samples for each treatment used for subsequent analysis were: 10 for D-W, 18 for W-D, and 13 for Sat. Filtering down to samples successfully analyzed for metabolites and 16S resulted in both data sources having the same set of 41 samples.

### 3.4. Predicting OTU Presence/Absence with Metabolomic Profiles

To test if metabolite abundance profiles could be used to predict whether an OTU would be observed (i.e., have a nonzero count) or not within a given moisture treatment, we used the following approach: for each OTU, data were converted to binary presence/absence data for each sample, i.e., if an OTU was observed with a non-zero count, it was denoted as present and otherwise absent. A random forest model [23] was fit to the data for each OTU using the *randomForest* R package [37]. The binary outcome of interest was presence/absence of the OTU for each sample, and the normalized log_2_ metabolite abundance values were used as the explanatory variables. The random forest model uses bootstrap aggregating as a validation strategy to protect against overfitting and to fairly evaluate the model’s predictive ability.

OTUs that were present for all or nearly all samples, or were absent for nearly all samples, are not amenable to classification. Moreover, they are not biologically interesting for the purposes of understanding how metabolites might predict the presence/absence of OTUs for individual samples. Thus, we required an OTU to have a minimum of 3 samples where the OTU was observed and a minimum of 3 samples for which the OTU was not observed. The number of OTUs meeting the above requirements were: 4916 for the Sat, 2987 for the D-W, and 5531 for the W-D treatments.

For each random forest model, i.e., each OTU and treatment, the following information was obtained: (i) variable importance metric for each variable, i.e., a metabolite; (ii) predictions (present/absent) for each sample based on the metabolomic abundance profiles; (iii) overall model predictive efficacy. Variable importance metrics were calculated as the mean decrease in the Gini index [22], a metric that can be used across models utilizing the same set of predictors. 

We used the balanced accuracy metric (Powers, 2011), the mean of the true positive and true negative prediction rates, to summarize model efficacy for the different moisture treatments (Appendix A). Balanced accuracy is used when there is an imbalance in the number of positive and negative labels and is equivalent to accuracy when labels are balanced between the two groups [38]. We set a balanced accuracy threshold greater than or equal to 80% to predict the presence/absence of OTUs to identify models to further investigate.

## 4. Results and Discussion

We aimed to test whether microbial taxa enrichment under a given soil moisture treatment could be a predictable trait as a function of the observed metabolites, particularly those belonging to classes of complex polysaccharides, long-chain fatty acids, and nitrogenous compounds. Therefore, in this study, we tested for the predictive power of observed metabolites in the ecological context of bacterial population shifts as result of moisture perturbations in wetland soils. We used a random forest classifier model to assess the effectiveness of metabolite data in the prediction of the presence/absence of patterns of microbial taxa obtained from 16S rRNA sequencing. The advantages of this approach include the ability to examine and detect linear and nonlinear relationships between observed metabolites and taxa in a multivariate manner; random forest analyses provide both a metric of how well the presence/absence of an OTU can be predicted and how useful a metabolite is for such a prediction in the presence of other metabolites. It also allowed us to examine patterns across similar taxa, i.e., assuming functional traits, and to make comparisons across treatments. In addition to high predictive performance, random forest classifiers can reveal feature importance [39]. Here, we applied this approach in the context of how the soil bacterial community responds to fluctuating moisture regimes.

### 4.1. Shifts in Relative Abundances of Bacterial Taxa 

The 16S rRNA gene sequences retrieved from all samples corresponded to 32 bacterial phyla-, 129 family-, and 218 genus-level OTUs. Ordination analyses performed using all (Figure 1a) and the top 100 OTUs (Appendix A) demonstrated separation of samples by moisture treatments. The relative abundances (>0.5%) of phyla are shown in Figure 1b. Among the dominant phyla based on the mean relative abundance across the three treatments were Acidobacteria, Actinobacteria, Bacteroidetes, Gemmatimonadetes, Latescibacteria, Planctomycetes, Proteobacteria, Verrucomicrobia, and Firmicutes. In the Sat treatment, Acidobacteria, Actinobacteria, Planctomycetes, and Firmicutes were significantly lower in relative abundance while Latescibacteria was higher (*p-adj* < 0.05) relative to both the W-D and D-W treatments. Organisms in the phylum Latescibacteria are capable of fermentative pathways as well as of oxygen-dependent metabolic reactions and have been recovered from oxygenated and partially/seasonally oxygenated aquatic habitats [40]. Their higher abundance in the Sat treatment is potentially due to this metabolic specialization compared to the other phyla, which are more physiologically diverse and ubiquitous in soils.

Relative shifts in class abundances in response to the moisture fluctuations may be reflective of ecological selection based on their primary respiratory pathways, i.e., aerobic or anaerobic. Acidobacteria Gp2, Acidobateria Gp7, and Betaproteobacteria were higher in the Sat treatment compared to those in W-D and D-W (Figure 1c). 

Acidobacteria_Gp1, Clostridia (obligate anaerobes), Deltaproteobacteria (aerobic), Gemmatimonadetes (both aerobic and anaerobic), and Sphingobacteria decreased in abundance in response to drying in the W-D samples while correspondingly increasing in response to wetting in the D-W samples. Spartobacteria and Alphaproteobacteria were higher in the D-W samples compared to the Sat samples. The Gemmatimonadetes are found in a wide range of environments and can adapt to low-moisture conditions [41]. Spartobacteria have been observed in diverse ecological niches and display aerobic, facultative anaerobic, or obligate heterotrophic lifestyles [42]. They are capable of utilizing various C compounds, e.g., cellulose, xylan [43,44], and sugars [45] or methane [46]. We interpret the relative shifts in class abundances in response to the moisture fluctuations to be reflective of ecological selection based on their primary respiratory pathways, i.e., aerobic or anaerobic.

### 4.2. Metabolite Shifts in Response to Hydrologic Shifts

Drying generally increased abundances of amino acids compared to wetting (Figure 2a). The abundances of nine amino acids shifted in both the W-D and D-W treatments relative to the Sat condition. For example, L-glutamic acid (*p* = 0.015) and L-isoleucine (*p* = 0.038) decreased significantly in the D-W compared to the Sat treatment (Figure 2a). Glycine (*p* = 0.024) decreased while L-serine (*p* = 0.004) and L-threonine (*p* = 0.000) increased in W-D compared to Sat. 

*N*-methylalanine (or 2-aminoisobutyric acid) decreased in both W-D (*p* = 0.009) and D-W (*p* = 0.038) relative to Sat. Under conditions of a high ratio of organic C to nitrogen, heterotrophic microorganisms compete with primary producers for the available nitrogen [47]. If much of the nitrogen is present in organic form, it is likely that the ability to use organic nitrogenous compounds as nitrogen sources would confer some advantages to organisms with this ability over those lacking it. Such utilization of ethylamine, beta-alanine, and L-serine as nitrogen but not as sole C sources have been well established. The ability to use methylamines as nitrogen sources by bacteria but not as sources of C has been widely shown in natural habitats [48]. In addition, Methylotrophic bacteria able to utilize methylamines as sole C and energy sources first convert the methyl groups to formaldehyde, which is subsequently either assimilated or oxidized to CO_2_ [49,50].

L-pyroglutamic acid decreased in both D-W (*p* = 0.022) and W-D (*p* = 0.007) relative to Sat. Among the amino acids that significantly increased in abundance in W-D compared to D-W were L-glutamic acid (*p* = 0.045), L-leucine (*p* = 0.000), L-serine (*p* = 0.000), L-threonic acid (*p* = 0.037), and L-threonine (*p* = 0.000). Protein amino acids such as glutamic acid, serine, and glycine can play significant roles as constitutive osmolytes or precursors thereof under soil drought [51,52]. The higher abundances of L-serine, L-leucine, L-threonic acid/threonine, and glutamic acid in the W-D treatment could be a stress response or simply accumulation of these small molecules in the soil exometabolome.

Higher relative abundances of methylalanine and glycine in the Sat treatment could indicate stimulation of pathways related to methylamine synthesis or degradation. Methylamines are produced from glycine and other amino acids (from protein or peptidoglycan), compatible solutes, and lipids. Expression of glycine-cleavage transcripts in methanogenic environments [53,54] and methanogenesis from methylamines by methanogens [55] and methylotrophs [56] have only been recently reported. Therefore, the higher abundance of methylalanine indicates conditions conducive to methanogenesis in the Sat treatment.

Organic acids such as benzoic acid, phenylacetate, and capric and lauric acid have been observed in anaerobic systems and have also been implicated in the methanogenic degradation of lignin derivatives [57]. Thirteen of the measured organic acids decreased in abundance in both D-W and W-D when compared to Sat (Figure 2b). Several organic acids increased in W-D compared to D-W, including those of 2-ketoisocaproic acid (*p* = 0.05), 3-hydroxybutyric acid (*p* = 0.048), DL-glyceraldehyde (*p* = 0.020), glycosylglyceric acid (*p* = 0.006), and methylsuccinic acid (*p* = 0.035). The organic acids that decreased in W-D compared to D-W included 3,4-hydroxybenzoic acid (*p* = 0.019), 4-hydroxycinnamic acid (*p* = 0.000), benzoic acid (*p* = 0.006), lauric acid (*p* = 0001), myristic acid (*p* = 0.005), and nonanoic acid (*p* = 0.018). A previously reported accumulation of short-chain fatty acids such as formate, isobutyrate, isovalerate, propionate, and pyruvate in the W-D samples occurred during the antecedent wet phase when anaerobic processes were dominant [22]. Therefore, the redox-active metabolites from fermentation and homoacetogenesis could be the precursors for active methanogenesis. 

### 4.3. Metabolites Predict Taxa

We rank metabolites based on their variable importance metric and present those found to be the strongest predictors of OTUs under each moisture condition (Figure 3). These were metabolites that had a variable importance score in the top 20% of variable importance scores across all models. Among these 34 top ranked metabolites, 24 were organic acids, 4 were amino acids, 3 were sugars, 1 was a carbonate ion, 1 was inositol, and 1 was uracil. Collectively, these metabolites were among the strongest predictors of 31 OTUs for the Sat treatment and 29 OTUs for the W-D treatment. However, we observed strong variabilities in the strength of metabolite–OTU predictability as a function of the moisture condition in question. For example, 3,4,5-trihydroxypentanoic acid, capric acid, and glycolic acid were among the strong predictors of the observed OTUs under the Sat condition. In contrast, 4-guanidinobutyric acid, 4 hydroxy-3-methoxybenzoic acid, 5 aminovaleric acid, D-xylose, glucosylglyceric acid, L-phenylalanine, maltotriose, sucrose, and toluic acid were strong predictors of OTUs under the W-D condition. Surprisingly, none of the metabolites had sufficient predictive power for the OTUs observed under the D-W condition (Appendix A).

In the W-D treatment, 4-guanidinobutyric acid, 4-hydroxy-3-methoxybenzoic acid, and glucosylglyceric acid strongly predicted the presence of Acidobacteria (Gp1, Gp10, Gp12, Gp5, and Gp6), Armatimonadetes, Planctomycetes, Deltaproteobacteria, and Subdivision3. D-xylose was a strong predictor of Parcubacteria, while sucrose was the strongest predictor of Acidobacteria_Gp5 among others from this phylum. Succinic acid also strongly predicted the presence of Parcubacteria and Firmicutes under the W-D condition and that of Acidobacteria_Gp22 and Gp18 under the Sat condition. Toluic acid ranked among the strongest predictors of Proteobacteria (Betaproteobacteria), Chlamydiae, Armatimonadetes, and Acidobacteria (Gp12, Gp10, and Gp1).

Our results show the high predictive power of D-xylose and sucrose for several members of the phylum Acidobacteria under the W-D treatment. Acidobacteria is a highly diverse phylum and is capable of metabolizing a wide range of carbohydrates as C sources, ranging from simple sugars to complex substrates such as D-glucose, D-xylose, and other polysaccharides [58]. Maltotriose, another carbohydrate, was also among the strong predictors of Acidobacteria although its metabolism is not well documented. These observations are consistent with previous studies that showed positive correlations between Acidobacteria subgroups and gene families related to C degradation, specifically those involved in hemicellulose degradation. For example, subgroup Gp4 has been described as the most versatile in the use of C from sucrose, maltose, and chitins [59,60]. It is important to note that much of our understanding of C degradation by Acidobacteria in soil is limited to correlative studies. The lack of isolates of most Acidobacteria subgroups is a challenge due to discrepancies between information from genomes of cultured representatives and that from marker gene assays.

In the Sat treatment, capric acid was consistently a strong predictor of Verrucomicrobia (Subdivision 5), Proteobacteria (betaproteobacteria), Firmicutes (Negativicutes), Chlamydiae (Chlamydiia), and Acidobacteria Gp6. Myo-inositol (a polyol) also had sufficient predictive power for many taxa including Verrucomicrobia Subdivision 5, Betaproteobacteria, Chloroflexi (Anaerolineae), Chlamydiae, and Acidobacteria Gp22. 3-aminoisobutyric acid and N-methylalanine, the methylamines that occurred in higher abundances in the Sat treatment, did not rank very high in predictive power for most OTUs, but in the W-D treatment were strong predictors of Firmicutes (Clostridia), Chlamydiae, and Acidobacteria_Gp18. None of the other amino acids ranked high enough based on our criterion for metabolite of predictive importance, although their relative abundances shifted significantly with wetting and drying. Glucosyslglyceric acid, significantly increased in abundance in the W-D treatment, has characteristics of compatible solutes under combined salt stress and nitrogen-limiting conditions and has been observed in prokaryotes across a wide taxonomic distribution [61]. Enrichment of the Latescibacteria in Sat treatment and their high predictability by capric acid, benzeneacetic acid, glycolic acid, polyols such as inositol and myo-inositols, and myristic acid support the diverse metabolic capacities of this phylum as also corroborated by recent reports of their biogeography [62].

The conversion of CO_2_ back to acetate by homoacetogenesis is a strictly anaerobic process; homoacetogenesis was shown to be preferred over methanogenesis in profundal sediments of Lake Constance [63], and anaerobic peatland soils [64,65] notwithstanding the predominance in low-temperature ecosystems. The higher relative abundances of Holophagae in the D-W and Sat treatments (Figure 1c) might be an indicator of homoacetogenesis as an active process that could potentially outcompete methanogenesis, as previously reported in swamp peats at pH 6.5 [64]. One member of the Holophagales order, *Holophoga foetida*, can also degrade aromatic compounds [66], an observation supported by our measurements of higher abundances of lactic acid, hydroxyphenylacetic, and 3-hydroxybutyric and methylsuccinic acids under saturated conditions (Figure 2b). All these acids have been implicated in anaerobic lignin degradation pathways [57]. Based on these observations, we hypothesize that homoacetogenesis is an active pathway in these soil systems. 

The presence of *Clostridia* under the wet and saturated conditions may explain the high levels of isopropanol and acetone accumulation reported previously [22]. The metabolic pathway of *Clostridia* remains poorly understood but recent studies suggest they may use pyruvate for biosynthetic reactions [67]. Elevated concentrations of acetate or butyrate, or both as we previously observed [22], have been suggested to trigger acetone production in *C. acetobutylicum* [68]. Close interactions between *Methanosarcina* and cellulolytic *Clostridia* have also been reported in freshwater methanogenic environments [69]. The combined observations of high concentrations of acetate and butyrate [22] and the strong predictive power of 3-aminoisobutyric acid from this study may support the co-occurrence of methanogenesis and fermentation processes upon rewetting soils.

The spatiotemporal variability in hydrological fluctuations in wetlands causes the development of temporal and spatial redox gradients [70,71,72,73], such that the subsequent manifestation of microbial processes is a function of the availability of terminal electron acceptors and metabolic substrates and products. Here, we provide evidence for shifts in microbial taxa and metabolite abundance in response to moisture shifts as a proxy for redox. Specifically, we find enrichment of drought-tolerant microbial taxa and sugars as metabolites upon drying a previously saturated soil; in addition, an increasing abundance of fatty acids with high predictive power for anaerobes such as *Clostridia*, *Holophagales*, and *Latescibacteria* under saturated conditions suggest a predominance of anaerobic C cycling. We, therefore, infer that present soil redox status may enrich for metabolite classes linked to anaerobic or aerobic metabolism, which, in turn, may predict the dominant microbial community structure in tidal wetlands under fluctuating moisture conditions. 

## 5. Conclusions

Soil moisture legacy and redox metabolism are intimately linked, yet it is difficult to discern how these soil conditions drive microbiome shifts within a predictive framework. Our results illustrate the potential of untargeted statistical approaches such as Random Forest models to interrogate complex soil microbiomes such as wetland sediments using multidimensional data such as metabolomics and the 16S rRNA gene-based community structure. The predictive power to consider discrete datasets for microbial population and/or substrate characteristics for wetland biogeochemistry, collectively and in relation, has previously been limited to correlation analyses. Elucidation of the chemical composition of microbial substrates such as fatty acids, polysaccharides and sugars, and amino acids and their dynamics in response to environmental factors is a critical step toward a predictive understanding of the complex relationship between soil chemical environment and microbial communities. Here, we demonstrate that taking a multidimensional data analysis approach to the interpretation of metabolomics has the potential to unravel the causative pathways selecting for the observed community structure. For example, the dominance of sugars and osmolytes in response to drought stress and that of organic acids under saturation conditions provide evidence of the dynamic nature of metabolome shifts with moisture while enriching for the specialized or facultative microbial taxa. The advantage of our approach is that the random forest model does not make any assumptions about the distribution (e.g., normal distribution) of variable values and can detect nonlinear relationships between predictors and response variables, unlike stepwise regression or correlation-based models that can only detect linear relationships. Such a linear relationship is not relevant to a complex microbial community structure.

## Figures and Tables

**Figure 1 microorganisms-10-01653-f001:**
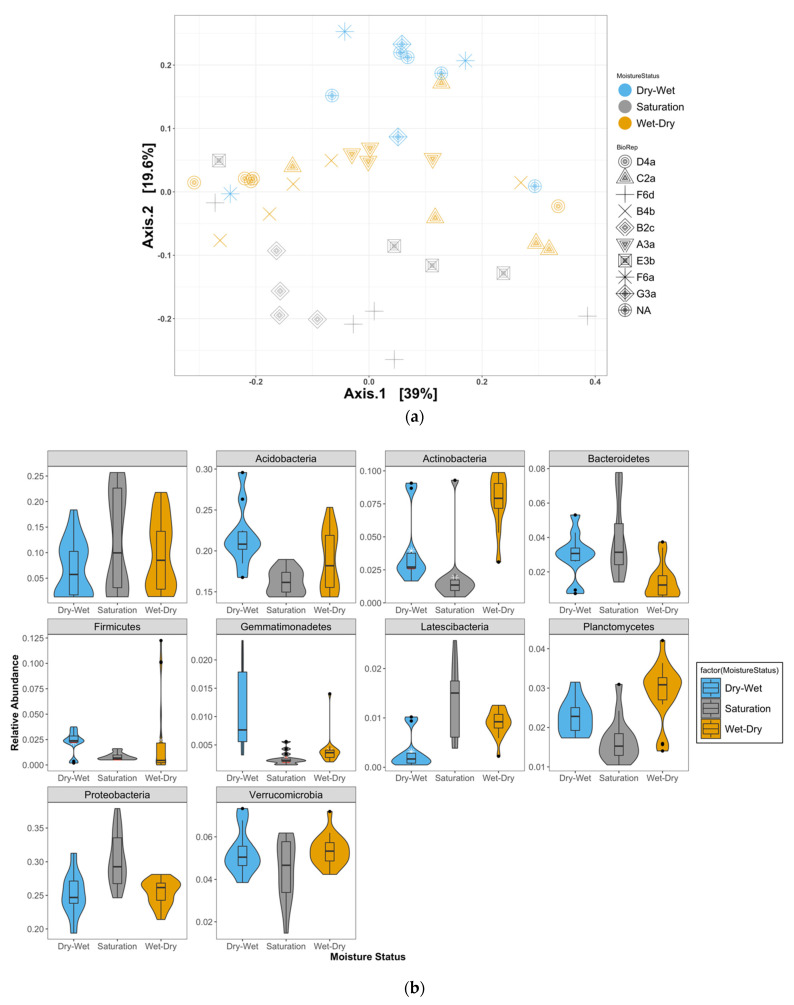
(**a**) Ordination plots showing separation of all OTUs by moisture treatments visualized by nonmetric multidimensional scaling (NMDS) of Bray–Curtis distance metrics. The three moisture treatments are shown in blue (Dry-Wet), grey (Saturation), and orange (Wet-Dry). The biological replicates and their technical replicates are shown by distinct symbols for each moisture treatment and listed under BioRep for their unique experimental IDs. (**b**) Relative abundance (>0.5%) of OTUs at the phylum level measured for each moisture treatment (Dry-Wet, Wet-Dry, and Saturation). Data are shown as violin plots where whiskers indicate the most extreme values within 1.5 multiplied by the interquartile region. Box, 25% quartile; median, 75% quartile. Pairwise comparisons of means to test treatment effects were performed after outlier removal. (**c**) Relative abundance (>0.5%) of OTUs at the class level measured for each moisture treatment (Wet-Dry, Dry-Wet, and Saturation). Data are shown as violin plots where whiskers indicate the most extreme values within 1.5 multiplied by the interquartile region. Box, 25% quartile; median, 75% quartile. Pairwise comparisons of means to test treatment effects were performed after outlier removal.

**Figure 2 microorganisms-10-01653-f002:**
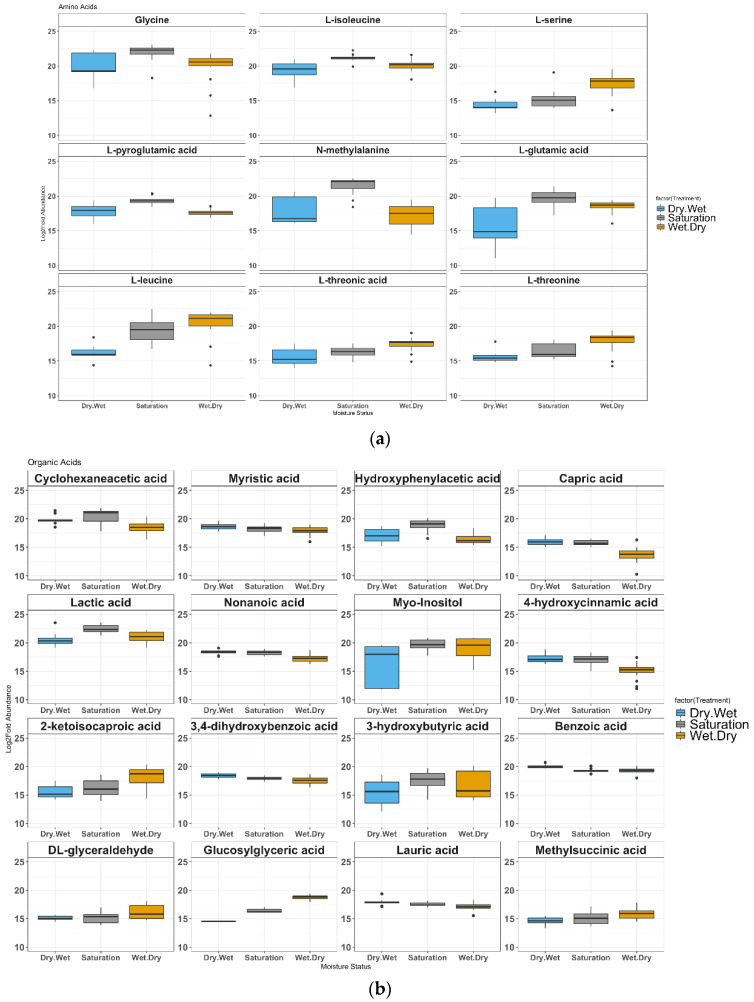
(**a**) Impact of soil moisture perturbations on metabolites shown as log2-fold abundances of amino acid that shifted significantly in response to a moisture treatment (Dry-Wet, Saturation, or Wet-Dry). Nine amino acids changed significantly under the wet or dry treatments compared to control (*p* < 0.05, one-way ANOVA). Whiskers indicate the most extreme values within 1.5 multiplied by the interquartile region. Box, 25% quartile; median, 75% quartile. Pairwise comparisons of means to test treatment effects were performed after outlier removal. (**b**) Impact of soil moisture perturbations on the soil metabolome shown as log2-fold abundances of metabolites (other than amino acids) that shifted significantly in response to a moisture treatment (Dry-Wet, Saturation, or Wet-Dry). Fifteen of 70 detected metabolites changed significantly under the wet or dry treatments compared to control (*p* <  0.05, one-way ANOVA). Whiskers indicate the most extreme values within 1.5 multiplied by the interquartile region. Box, 25% quartile; median, 75% quartile. Pairwise comparisons of means to test treatment effects were performed after outlier removal.

**Figure 3 microorganisms-10-01653-f003:**
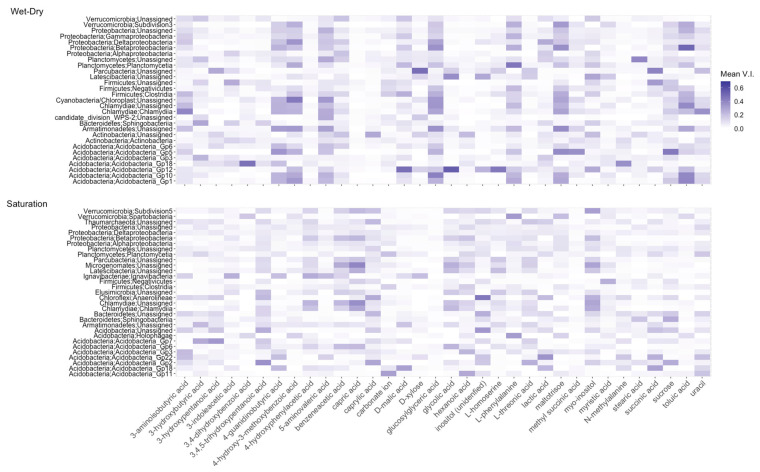
Metabolites ranked based on their mean variable importance for predicting OTUs under Wet-Dry (**top**) and Saturation (**bottom**) treatments.

## Data Availability

The data presented in this study are available on request from the corresponding author.

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
