# Peer review of "Soil Metabolomics Predict Microbial Taxa as Biomarkers of Moisture Status in Soils from a Tidal Wetland"

_microorganisms, 2022, doi:10.3390/microorganisms10081653_

Round 1
Reviewer 1 Report
This is an excellent paper, the only drawback was that the number of replicates sometimes was not enough in case of metabolites detection, but the authors properly treat this by using adequate statistical approaches.
Only one question: What GC-MS systems was used for metabolites detection?
Author Response
Thank you very much for your favorable comments and encouraging words for our work.
We have now included the details for the GC-MS used in lines: 211-213.
Reviewer 2 Report
The present study shows a current topic but from a very novel perspective, which may be very useful in the future. Omics are increasingly popular and good management of the huge amount of data they provide can result in valuable information that sometimes escapes us. The whole text is well presented and discussed. However, I suggest some modifications to improve the scientific quality of the manuscript. The recommendations are as follows:
- Complete the authors' addresses.
- Introduction part. The statement and hypothesis carried out in the manuscript should be at the end of the introduction. Therefore, I suggest combining all the research related to this manuscript in the same paragraph (lines 72-74, 84-87, 88-101).
- When was the experiment performed?
- Ln 424-436. This part is hard to read and understand. Please, reduce the description of the significant differences observed.
- Figure captions. Please, include what is represented in each figure, i.e. median or mean, standard error or standard deviation...
- Revise the whole text and erase the double spaces and the spaces missed (Ln 525: "...a acids..."
